# Lower Limb Motion Recognition Based on sEMG and CNN-TL Fusion Model

**DOI:** 10.3390/s24217087

**Published:** 2024-11-04

**Authors:** Zhiwei Zhou, Qing Tao, Na Su, Jingxuan Liu, Qingzheng Chen, Bowen Li

**Affiliations:** 1College of Intelligent Manufacturing Modern Industry, Xinjiang University, Urumqi 830017, China; zzw912918131@163.com (Z.Z.); awbfineyd@gmail.com (N.S.); 13659939770@163.com (J.L.); chenqingzheng@stu.xju.edu.cn (Q.C.); lbw04280@163.com (B.L.); 2The First Affiliated Hospital, Xinjiang Medical University, Urumqi 830017, China

**Keywords:** surface electromyography signals, lower limb action recognition, convolutional neural network, transformer encoder, long short-term memory

## Abstract

To enhance the classification accuracy of lower limb movements, a fusion recognition model integrating a surface electromyography (sEMG)-based convolutional neural network, transformer encoder, and long short-term memory network (CNN-Transformer-LSTM, CNN-TL) was proposed in this study. By combining these advanced techniques, significant improvements in movement classification were achieved. Firstly, sEMG data were collected from 20 subjects as they performed four distinct gait movements: walking upstairs, walking downstairs, walking on a level surface, and squatting. Subsequently, the gathered sEMG data underwent preprocessing, with features extracted from both the time domain and frequency domain. These features were then used as inputs for the machine learning recognition model. Finally, based on the preprocessed sEMG data, the CNN-TL lower limb action recognition model was constructed. The performance of CNN-TL was then compared with that of the CNN, LSTM, and SVM models. The results demonstrated that the accuracy of the CNN-TL model in lower limb action recognition was 3.76%, 5.92%, and 14.92% higher than that of the CNN-LSTM, CNN, and SVM models, respectively, thereby proving its superior classification performance. An effective scheme for improving lower limb motor function in rehabilitation and assistance devices was thus provided.

## 1. Introduction

The recognition of lower limb motion holds significant application value across various fields. In rehabilitation treatment, it allows doctors to objectively assess a patient’s physical condition, thereby optimizing the rehabilitation plan. Additionally, lower limb motion recognition facilitates the effective integration of the human body with wearable power-assisted exoskeleton robots, enhancing both weight-bearing capacity and work efficiency. It is also reported that identifying lower limb movements improves the quality of daily life in elderly, frail, and motor-impaired patients. Composed of superimposed action potentials generated by multiple motor units, sEMG contains biological information closely related to human behavior. The acquisition of sEMG signals, when compared to intramuscular EMG signal acquisition, offers significant advantages, including high safety, strong reliability, and non-invasiveness [1]. Widely used in rehabilitation robots, wearable exoskeleton control, and human–computer interactions, surface electromyography signals directly reflect muscle activity patterns.

The preprocessing research of surface electromyography (sEMG) and dynamic electroencephalogram (EEG) signals primarily focuses on feature extraction methods and classification algorithms, with the aim of achieving a high recognition rate for various human limb movement patterns [2]. For example, Totty et al. effectively classify daily life activities in the functional arm activity behavior observation system by adopting the K-nearest neighbor (KNN) algorithm combined with muscle activation and movement data [3]. Wang et al. analyze the shortcomings of the traditional threshold function in sEMG signal denoising and propose an improved threshold function. The classification performance of LSTM is then compared with that of CNN, SVM, and other classification algorithms [4]. Cui et al. combine the sEMG feature space construction method using time-domain and time–frequency-domain features and propose an SVM classifier based on sEMG, an AFSA-optimized SVM classifier, and a deep learning CNN classification model to recognize upper limb gestures [5]. Shi proposes a lower limb rehabilitation training mode based on surface electromyography (sEMG) signals. By extracting features from the sEMG signal and using a BP neural network to identify movement intention, the result is then used as the driving signal for the rehabilitation robot to facilitate active training [6]. Zheng et al. establish a mapping model between sEMG signals and motion angles by extracting features and utilizing a back propagation neural network. This approach predicts joint single-degree-of-freedom motions and combined motion angles with high accuracy [7]. Huang et al. propose an improved deep forest model for hand action recognition using sEMG data. By leveraging the complementarity between three classifier algorithms, they enhance the classification and recognition accuracy of the deep forest model, achieving an average recognition accuracy of 94% for 16 commonly used hand actions [8]. Ai et al. extract time-domain features and wavelet coefficients from surface EMG signals and utilize dynamic time warping (DTW) distance to extract features from acceleration signals. They then apply linear discriminant analysis (LDA) and SVM to classify five lower limb movements [9]. Liu et al. utilize the improved RelieF algorithm to select the optimal time-domain feature vector after data preprocessing and apply a support vector machine based on a balanced decision tree to layer the identification of five common dumbbell actions. This approach lays the foundation for personalized dumbbell action guidance [10]. Zhong et al. propose a multi-scale time-frequency information fusion representation method (MTFIFR) to obtain time-frequency features of multi-channel sEMG signals. They design a multi-feature fusion network (MFFN) and introduce a deep belief network (DBN) as the classification model for MFFN, aiming to improve the generalization performance for a broader range of upper limb movements [11]. Zhao et al. adopt a data-enhanced EFM as the input sample and integrate an ECA mechanism into the CNN architecture to assign higher weights to key feature information. Through modular design, they further adjust the layers of the deep feature extraction module within the network, effectively improving the accuracy of gesture recognition [12]. Xi et al. apply wavelet transform to process sEMG signals, calculate wavelet coherence coefficients, and use SVM to classify six types of daily activities [13]. Wei et al. proposed a multi-stream convolutional neural network (CNN) framework to improve the recognition accuracy of gestures by using a “divide and conquer” strategy to learn the correlation between a single muscle and a specific gesture [14]. Liu et al. propose a muscle fatigue classification method based on electromyography signals, which incorporates the crossover mutation of a genetic algorithm and the improved fruit fly optimization algorithm. This approach is combined with a neural network to identify muscle fatigue, enabling accurate detection and classification of muscle fatigue [15]. Sui et al. decompose the sEMG signal using wavelet packet transform (WPT) to extract wavelet packet coefficients and feed the calculated variance and energy as inputs to the improved SVM classifier, achieving an accuracy of 90.66% in recognizing six upper limb actions [16]. Liu et al. improve the accuracy of hand movement intention classification by enhancing the activity segment detection technology of sEMG. They employ a two-stage discriminative adaptive threshold technique to detect the active segment of sEMG, using the feature matrix corresponding to the motion intention and its label as the input and output of the LSTM hand motion intention classification model. This approach increases the average classification accuracy of six motion intentions to 91.7% [17]. Gupta et al. record dual-channel EMG signals from 18 subjects across three lower limb motion modes and evaluate the influence of window size, feature vector type, and classifier type on recognition performance. They find that selecting a 256 ms window size, 32 ms overlap, LDA classifier, and temporal feature vector yields excellent performance [18].

According to Refs. [19,20], current motion recognition technology primarily relies on EMG and EEG signals. The use of surface EMG signals for lower limb action recognition achieves satisfactory results under objective conditions. Existing research on lower limb action classification methods generally focuses on extracting sEMG features, which are then identified and classified using machine learning or deep learning algorithms to recognize lower limb actions. However, this approach to manual feature extraction addresses only a segment of the information, failing to encapsulate the complete characteristics of the input data and missing essential features. As a result, certain sEMG information is lost in the feature extraction process, which adversely affects the classification accuracy. To address this issue of information loss in traditional feature extraction, this study proposes a CNN-TL lower limb motion recognition model based on surface EMG signals. The model effectively captures spatio-temporal features in EMG signals, enhancing the recognition accuracy and robustness of lower limb actions, and provides a more reliable and efficient solution for lower limb motion rehabilitation.

## 2. Materials and Methods

### 2.1. Experimental Objects and Equipment

In this study, sEMG data from 20 healthy subjects are collected, with the basic body parameters displayed in Table 1. None of the subjects suffer from a fracture, sprain, muscle strain, or any other injury that could affect motor function prior to the start of the experiment. No strenuous exercise is performed one week before the experiment, effectively avoiding muscle soreness or discomfort. The PLUX wireless EMG acquisition device is used as the experimental equipment, with a sampling frequency of 1000 Hz selected. Additionally, silver chloride (AgCl) electromyography electrode sheets and 75% alcohol wipes are used.

### 2.2. Experiment Procedure

The experiment is conducted in the Intelligent Medical Rehabilitation Robotics Laboratory. To reduce impedance, the hair on the skin surface of the subject’s target areas is removed before the experiment, and the area is wiped with 75% alcohol to remove surface oils. After the alcohol dries, electrodes are placed at the highest point of the target muscle bulge, ensuring consistent spacing between electrodes. In this study, selected muscles for analysis include the rectus femoris, vastus lateralis, vastus medialis, semitendinosus, tibialis anterior, lateral gastrocnemius, and medial gastrocnemius. These muscles are involved in coordinated force generation during walking, stair ascent and descent, and squatting. By selecting these muscles, comprehensive capture of muscle activation patterns during common lower limb movements is achieved, providing support for targeted rehabilitation planning, ensuring coordinated muscle development, and effectively guiding movement quality and functional recovery, as shown in Table 2:

As shown in Figure 1, subjects perform each of the four lower limb movements: ascending stairs, squatting, walking, and descending stairs. For the two lower limb movement modes of ascending and descending stairs, each subject performs 8 sets of each movement, with each set requiring 5 repetitions, and each movement cycle lasting 2 s. Each subject performs three sets of 20 squats, with each exercise cycle lasting 3 s. Five sets of walking exercises are performed, with each exercise cycle lasting 60 s, yielding 30 sets of data. A rest period of 3–5 min is taken between sets, and 10–15 min of rest is observed between different movements to avoid muscle fatigue affecting the results.

### 2.3. Data Preprocessing

During sEMG acquisition, environmental noise and other bioelectric signals, such as ECG signals, can easily interfere with the results, making noise reduction essential, as shown in Figure 2. The frequency range containing useful information in sEMG is primarily concentrated between 5 and 200 Hz. Therefore, in this study, a fourth−order Butterworth bandpass filter ranging from 30 to 300 Hz, along with a 50 Hz notch filter, is applied to filter the raw sEMG signals. Additionally, the db2 wavelet basis is used for 4−level wavelet decomposition, and the wavelet threshold technique is employed to further reduce noise.

The processed data are subsequently segmented according to the initial point of each action cycle, with the action cycles for walking and ascending and descending stairs set to 2 s, and the cycle for squatting set to 3 s. The overlapping window technique is subsequently employed to further divide the sEMG data within each action cycle. A sliding window of 1024 ms with a step size of 512 ms is utilized to break down the sEMG from each channel into serialized data windows. After expanding the data from 7 channels using this sliding time window, 8640 signal data samples, each of size 7 × 1024, are obtained. Figure 3 illustrates the process of segmenting EMG signals from a single channel using sliding window technology. After screening, the final number of samples obtained is as follows: 2160 samples for ascending stairs, 1920 samples for squatting, 1680 samples for walking, and 2160 samples for descending stairs.

### 2.4. Feature Extraction and Analysis

To extract effective information from the processed EMG data for use as feature input in the machine learning model, this study extracts both time-domain and frequency-domain features from the collected EMG signals and normalizes these features. In the time-domain analysis, key metrics such as the mean absolute value (MAV) and root mean square (RMS) of the EMG time series $x_{i}$ are considered. For frequency-domain analysis, mean power frequency (MPF) and median frequency (MF) are further extracted as essential frequency-domain features by analyzing the power spectral density function P(f) of the EMG signals. The calculation formulas are presented in Table 3.

### 2.5. Lower Limb Action Recognition Model

#### 2.5.1. CNN

sEMG is considered a mixed signal, originating from the temporal and spatial superposition of multiple muscle activities, leading to its high complexity [21]. Traditional convolutional neural networks, primarily used for image processing, struggle to effectively capture features in one-dimensional time series data. In contrast, a one-dimensional convolutional neural network (1DCNN) directly handles time series data, allowing the model to have fewer parameters while effectively learning features from these sequences. Therefore, this paper selects 1 1DCNN as the model to process sEMG, providing robust support for action recognition. The model architecture is illustrated in Figure 4.

In this paper, a 1DCNN model is employed, consisting of three convolutional layers, pooling layers, batch normalization (BN) layers, three fully connected layers, and a Dropout layer. The input signal in the convolutional layer is processed through one-dimensional convolution and calculated as follows:(1)$$h_{j}^{l}=\sum_{i=1}^{n}x_{i}^{l}⊗k_{ij}^{l}+b_{j}^{l}$$

In the formula, *l* represents the number of layers, *j* denotes the ordinal number of elements, *n* is the length of the input feature vector, *h* is the output feature vector, *x* represents the input feature vector, *k* is the convolution kernel, and *b* is the bias vector.

To enhance the training efficiency and stability of the model, batch normalization (BN) is applied after each convolutional layer. The calculation process is as follows:

First, the mean and variance of the same channels are calculated along the batch dimension, with the calculation expressed by the following formula:(2)$$\mu_{ß}=\frac{1}{n}\sum_{i=1}^{n}x_{i}$$
(3)$$\sigma_{ß}^{2}=\frac{1}{n}\sum_{i=1}^{n}{\left(x_{i}-\mu_{ß}\right)}^{2}$$

Then, the mean and variance along the same batch dimension are normalized, with the calculation expressed by the following formula:(4)$${\hat{x}}_{i}=\frac{x_{i}-\mu_{ß}}{\sqrt{\sigma_{ß}^{2}+\varepsilon }}$$

Next, any necessary scaling and shifting to restore the eigenvalue are performed by learning two parameters, γ (scaling factor) and β (translation factor), calculated as follows:(5)$$y_{i}=\gamma {\hat{x}}_{i}+\beta \equiv {\mathrm{BN}}_{\gamma ,\beta }(x_{i})y$$

Finally, the *ReLU* activation function is applied after each BN, with the calculation expressed by the following formula:(6)$$ReLU(x)=\max(0,x)=\left\{\begin{array}{cc} x & x\geq 0\\ 0 & x<0 \end{array}\right.$$

To improve the model training speed, Max pooling is used for downsampling. During the training phase, the Cross Entropy Loss function is employed to evaluate the deviation between the probability distribution of the model output and the true label, calculated as follows:(7)$$L(\theta )=-\frac{1}{n}\left[\sum_{i=1}^{n}\sum_{k=1}^{K}I\{y_{i}=k\}\log\frac{\exp(\theta_{K}^{T}x)}{\sum_{j=1}^{K}\exp(\theta_{j}^{T}x)}\right]$$

Additionally, to optimize the learning process and minimize the loss function, the Adadelta optimizer is adopted in this study to automatically adjust the learning rate. Two Dropout layers are inserted between the three fully connected layers to enhance the model’s generalization ability and prevent overfitting. The network model configuration is shown in Table 4.

#### 2.5.2. LSTM

The long short-term memory (LSTM) network is a specialized recurrent neural network. By introducing a gating unit to control data flow, it effectively addresses the issues of gradient vanishing and gradient explosion commonly encountered by traditional recurrent neural networks when processing long sequence data [22]. EMG signals often contain complex temporal dynamics that may exhibit correlations over extended time scales, and the characteristic architecture of LSTMs enables them to learn these long-term dependencies. The structure of the LSTM network is shown in Figure 5.

The working mechanism of LSTM includes processing the current input $X_{t}$ and the hidden state $H_{t-1}$ from the previous time step through the sigmoid function to generate the forgetting gate output $F_{t}$, which varies between 0 and 1. An output of 0 indicates “completely forgotten”, while a value of 1 indicates “fully retained”. The calculation formula is as follows:(8)$$F_{t}=\sigma (X_{t}W_{xf}+H_{t-1}W_{hf}+b_{f})$$

In the formula, $W_{f}$ represents the weight, σ is the sigmoid activation function, and $b_{f}$ denotes the bias.

The input gate plays a crucial role in updating the cell state. First, the portion of information that needs to be updated is determined by the sigmoid function, generating the output $I_{t}$ of the input gate. Next, a new candidate value vector $\tilde{C_{t}}$ is generated through the tanh layer, representing new information that may be added to the state. Finally, the output of the forget gate $F_{t}$ is multiplied with the previous cell state $C_{t-1}$ to discard unnecessary information, while the output of the input gate $I_{t}$ is multiplied with the candidate value $\tilde{C_{t}}$ to update the new cell state $C_{t}$. The formula is given as follows:(9)$$I_{t}=\sigma (X_{t}W_{xi}+H_{t-1}W_{ht}+b_{i})$$
(10)$${\tilde{C}}_{t}=\tanh(X_{t}W_{xc}+H_{t-1}W_{hc}+b_{c})$$
(11)$$C_{t}=F_{t}⊙C_{t-1}+I_{t}⊙{\tilde{C}}_{t}$$

In the formula, $\tilde{C_{t}}$ represents candidate vectors, ${F_{t}*C}_{t-1}$ denotes the selective forgetting of irrelevant information, and $I_{t}*\tilde{C_{t}}$ signifies the retention of useful information.

In LSTM, the output gate processes the cell state through a sigmoid layer to determine what should be output. The cell state is then adjusted through a tanh layer and multiplied by the output of the sigmoid layer to produce the final output $H_{t}$. The formula is as follows:(12)$$O_{t}=\sigma (X_{t}W_{xo}+H_{t-1}W_{ho}+b_{o})$$
(13)$$H_{t}=O_{t}⊙\tanh(C_{t})$$

In the formula, $O_{t}$ represents the activation vector of the output gate.

The parameter configuration of the LSTM network used in this study is presented in Table 5.

#### 2.5.3. Transformer

The transformer structure includes an encoder and a decoder, with the encoder used for classification [23]. It primarily utilizes core components such as the multi-head attention mechanism, positional encoding, and feed-forward networks to capture long-term dependencies and complex signal patterns. The complete structure is illustrated in Figure 6.

(1) Positional encoding: In the transformer network, since the self-attention mechanism is used to extract information without a recursive structure, positional encodings are added to provide the model with information about the signal order. The formula is as follows:(14)$$PE_{\left(pos,2i\right)}=\sin(pos/10000^{2i/d})$$
(15)$$PE_{\left(pos,2i+1\right)}=\cos(pos/10000^{2i/d})$$

(2) The self-attention mechanism, the core of the transformer network, enables the model to weight each element based on its relationship to other elements in the sequence when processing sequential data. The formula is given as follows:(16)$$Attention\left(Q,K,V\right)=Softmax\left(QK^{T}/\sqrt{D_{K}}\right)V$$

In the formula, Q is the query matrix, K is the key matrix, V is the value matrix, and $D_{K}$ represents the matrix dimension. Transformer converts single-head self-attention to multi-head self-attention through concatenation, with the calculation expressed by the following formula:(17)$$MultiHead(Q,K,V)=Concat(head_{1};\cdots ;head_{h})W^{o}$$
(18)$$head_{i}=Soft\max(\frac{QW_{i}^{Q}{\left(KW_{i}^{K}\right)}^{T}}{\sqrt{d_{k}}})VW_{i}^{V}$$

In the formula, MultiHead represents the multi-head attention mechanism function, and ${head}_{i}$ denotes the output of the *i*th attention head.

(3) Layer normalization and residual connections: Layer normalization stabilizes training by standardizing input features [24]. Residual connections facilitate the direct flow of gradients through the network, helping to prevent the vanishing or exploding gradient problem in deep networks. The calculation formula is as follows:(19)$$X_{out}=LayerNorm=x_{ij}-\mu_{j}/\sqrt{\sigma_{j}^{2}+\varepsilon }$$

In the formula, $\mu_{j}$ represents the mean, and $\sigma_{j}^{2}$ denotes the variance.

#### 2.5.4. CLT Model

Building on the above analysis, this study constructs a CNN-Transformer-LSTM (CNN-TL) lower limb motion recognition model based on sEMG signals, with its overall framework illustrated in Figure 7. First, the preprocessed sEMG input is sent to the CNN layer for feature extraction. After passing through two fully connected layers, the data are successively processed by the Transformer and LSTM layers. Finally, the fully connected layer generates the final classification decision. For multi-channel EMG signals, CNN analyzes the interactions between different muscles and their spatial layout. The transformer effectively captures long-range dependencies within sequences through its self-attention mechanism, while LSTM handles time series data and retains long-term dependencies. By integrating these three models, sEMG signals are identified efficiently and accurately.

## 3. Results

### 3.1. Model Evaluation Index

In evaluating classification task performance, this paper adopts four core statistical measures as evaluation criteria: accuracy, precision, recall, and F1 score [25]. These metrics comprehensively reflect multiple dimensions of the classification model’s performance, providing a basis for the overall evaluation of the model. The calculation formulas are as follows:(20)$$Accurary=\frac{TP+TN}{TP+TN+FP+FN}$$
(21)$$Recall=\frac{TP}{TP+FN}$$
(22)$$Precision=\frac{TP}{TP+FP}$$
(23)$$F1Score=2\times \frac{Recall\times Precision}{Recall+Precision}$$

In the formula, TN represents the count of negative samples correctly identified as the negative class, TP represents the count of positive samples accurately classified as the positive class, FN denotes the number of cases where a positive sample is incorrectly classified as the negative class, and FP denotes the number of instances where a negative sample is incorrectly classified as the positive class.

### 3.2. Analysis of Experimental Results

During 300 rounds of training on the CNN-TL model, changes in the model’s loss value are tracked, and the fitting quality is evaluated by analyzing the training and test loss curves, as shown in Figure 8. The training results indicate that the model’s loss value is reduced to just 1.7%, while the recognition accuracy reaches 96.13%, further confirming the excellent convergence of the CNN-TL model.

Figure 9 presents a comparative analysis of the results from the proposed four classification model recognition schemes. For the nonlinear and multiclass challenges of sEMG feature vectors, a one-vs-all classification method is employed. Specifically, multiple four-class support vector machine (SVM) models are trained, with each model designating one class as the positive class and treating the others as negative classes. In the SVM models, the kernel selected is the Gaussian kernel, with the penalty parameter C set to 1 and the γ (gamma) value set to 0.1. The data reveal that the CNN-TL model demonstrates remarkable performance across various indicators. Specifically, the model achieves an accuracy of 96.13%, a precision of 95.71%, a recall of 95.60%, and an F1 score of 95.65%. Notably, the accuracy of the CNN-TL model is 3.76% higher than that of the CNN-LSTM model, 5.92% higher than that of the CNN model, and 14.92% higher than that of the SVM model. Compared to the CNN-LSTM model, the CNN-TL model’s precision, recall, and F1 score increase by 4.61%, 3.37%, and 3.99%, respectively. When compared to the CNN model, these increases are 6.77%, 8.64%, and 7.71%, respectively. Compared to the SVM model, the improvements are 13.33%, 16.63%, and 15.01%, respectively. These results demonstrate that the CNN-TL model exhibits higher adaptability and superiority in lower limb movement recognition than the CNN-LSTM, CNN, and SVM models.

To evaluate the performance of various lower limb motion recognition models, CNN-TL, CNN-LSTM, CNN, SVM, and other models are tested using the same dataset. When dealing with different lower limb movements, the probability of the test sample being classified into a specific category is first calculated. Based on these probabilities, the ratio of false positives and true positives under each threshold condition are determined, and the corresponding ROC curve is generated. For each lower limb movement, ROC curves are generated in each test set. By averaging the ROC curves generated by each action recognition model, comprehensive ROC curves for each model in the lower limb action recognition task are obtained, as shown in Figure 10.

As shown in Figure 11, the confusion matrix of the four different models displays the classification results for four lower limb movements, clearly demonstrating that the proposed CNN-TL model accurately identifies various lower limb movements based on the surface EMG signals from the seven channels of the lower limb. In the confusion matrix, the diagonal elements represent the quantity of samples correctly classified for each action, while the non-diagonal elements indicate the quantity of samples incorrectly classified. The categories are defined as follows: 0 for walking, 1 for ascending stairs, 2 for descending stairs, and 3 for squatting.

### 3.3. Ablation Experiment

In this section, to validate the effectiveness of the CNN-TL model, the influence of various components on model performance is examined. During the training process, the convolutional neural network (CNN) serves as the benchmark model to explore the effectiveness of the CNN-TL model and the specific influence of each component on its performance. The models are decomposed into CNN, CNN-LSTM, and CNN-Transformer for comparative analysis. The experimental results are presented in Table 6.

Analyzing the experimental data in Table 6 reveals that, compared to standalone CNN and LSTM models, the CNN-LSTM configuration exhibits superior performance. This improvement is attributed to the synergy between CNN and LSTM, enabling not only the extraction of spatio-temporal features but also the capture of dynamic data changes along the time dimension. The LSTM-Transformer fusion model combines LSTM’s proficiency in processing time-series data with transformer’s capability to capture long-range dependencies. Owing to its self-attention mechanism, the CNN-Transformer model demonstrates a natural aptitude for handling sequences of varying lengths, markedly surpassing the classification efficacy of singular models. Notably, the CNN-TL model achieves a classification accuracy of 96.13%, outperforming the other configurations. This superiority arises from the introduction of the Transformer, which effectively addresses long-term dependency challenges, while the LSTM component enhances the model’s comprehension of short-term dependencies in time series. This amalgamation not only optimizes the model’s processing of sequential data but also augments its ability to discern intrinsic data patterns from multiple dimensions, thereby enhancing its generalization performance on novel, previously unseen data.

## 4. Conclusions

Due to the CNN model’s restricted capacity to capture global information from extended series of EMG signals, the CNN-TL model is proposed by integrating Transformer and LSTM. This combined model effectively captures global information and sequence relationships, addressing the issue of missing local feature information in the CNN model. In this investigation, four distinct motion recognition models were developed and assessed to classify four specific movements during lower limb rehabilitation exercises. From the sEMG data collected from 20 participants, four key features—MAV, RMS, MPF, and MDF—were extracted and applied as inputs to the machine learning models. The preprocessed sEMG data were then used for the learning and training of the CNN, LSTM, CNN-LSTM, CNN-Transformer, and CNN-TL models. This study compares the performance of each model on the lower limb motion recognition task. The experimental results indicate that the CNN-TL model achieves an average accuracy of 96.13% on the test data.

The CNN-TL model proposed in this study demonstrates significant advantages over other models, confirming its excellent performance in the field of lower limb motion recognition. In future reseaarch, the study plans to expand the participant group from healthy individuals to patients with impaired lower limb function, further standardizing the EMG acquisition process to enhance the action recognition accuracy of the lower limb rehabilitation robot based on sEMG across different patient groups. Additionally, different training models will be explored and refined to optimize action recognition performance.

## Figures and Tables

**Figure 1 sensors-24-07087-f001:**
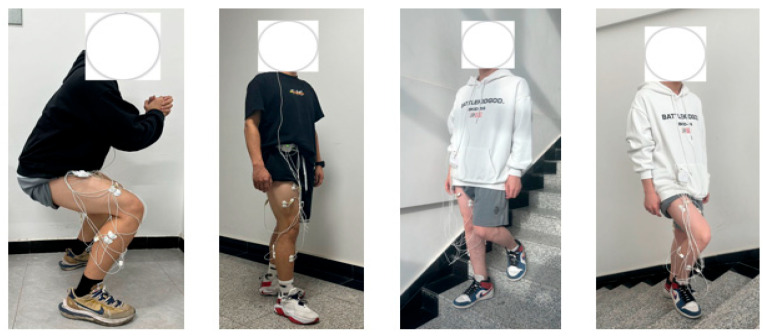
Diagram of the experimental scenario.

**Figure 2 sensors-24-07087-f002:**
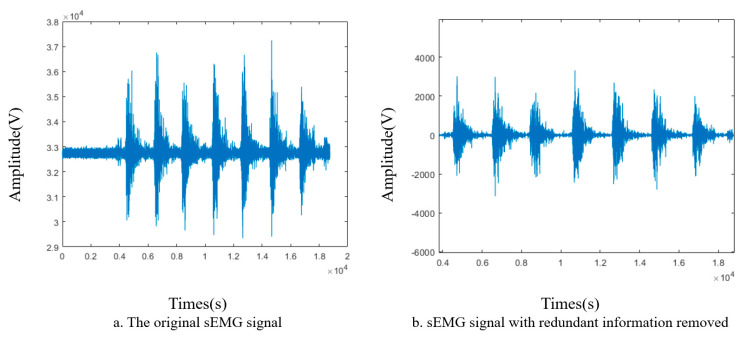
Elimination of extraneous information signals from sEMG.

**Figure 3 sensors-24-07087-f003:**
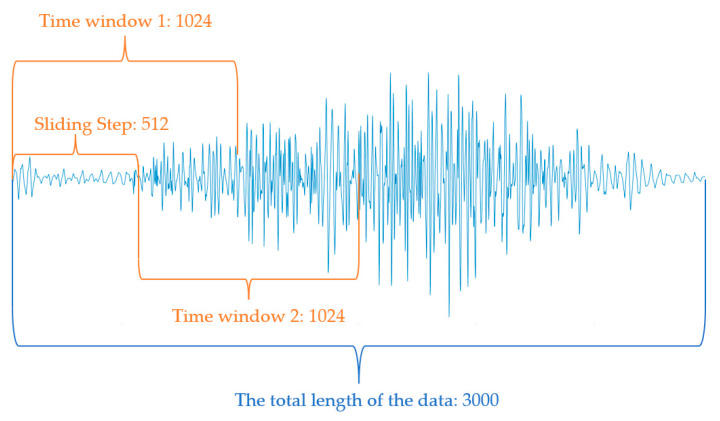
Sliding window segmentation diagram of single-channel sEMG signal.

**Figure 4 sensors-24-07087-f004:**
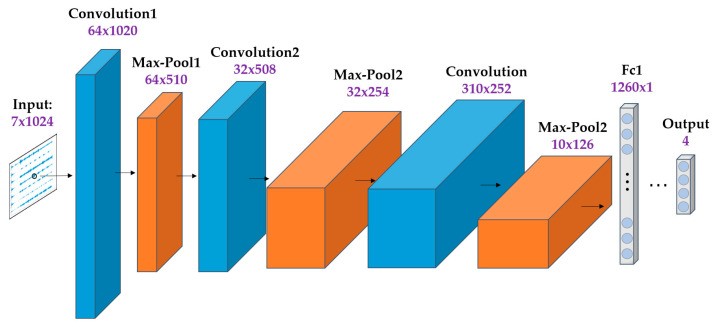
1DCNN model structure.

**Figure 5 sensors-24-07087-f005:**
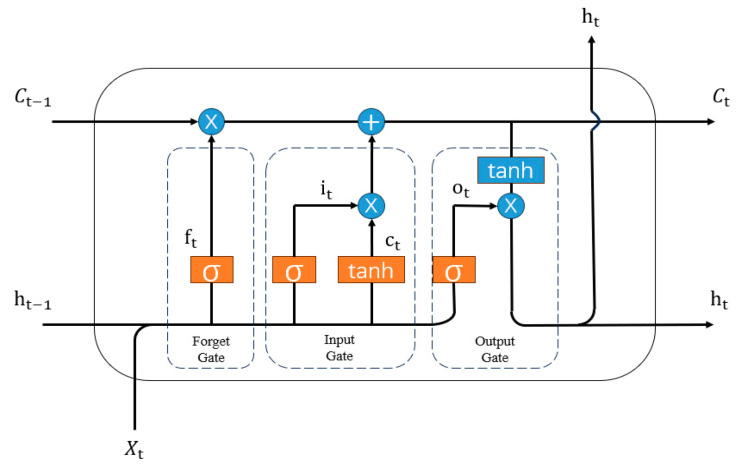
LSTM model structure.

**Figure 6 sensors-24-07087-f006:**
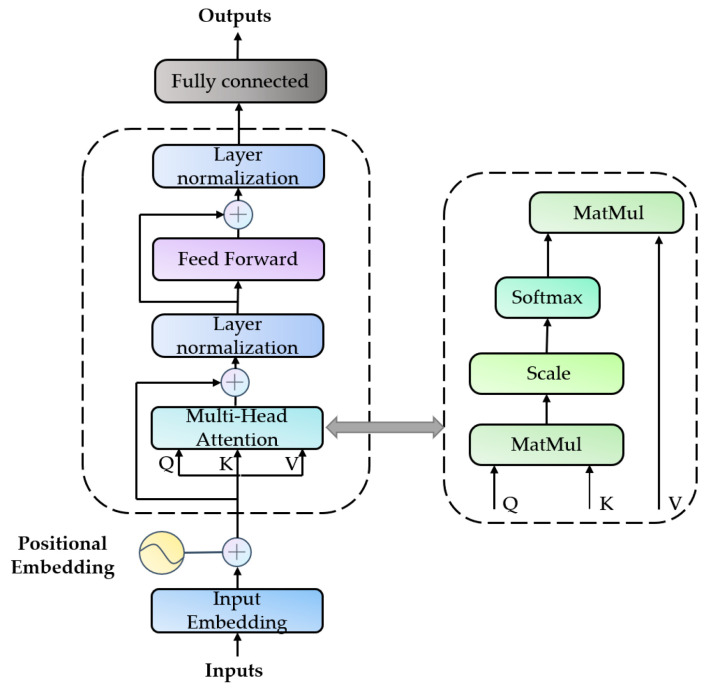
Transformer encoder model structure.

**Figure 7 sensors-24-07087-f007:**
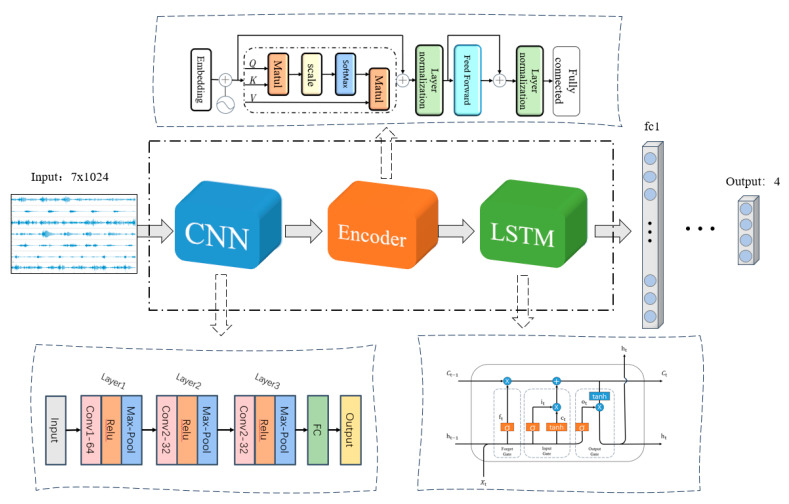
CNN-TL overall architecture.

**Figure 8 sensors-24-07087-f008:**
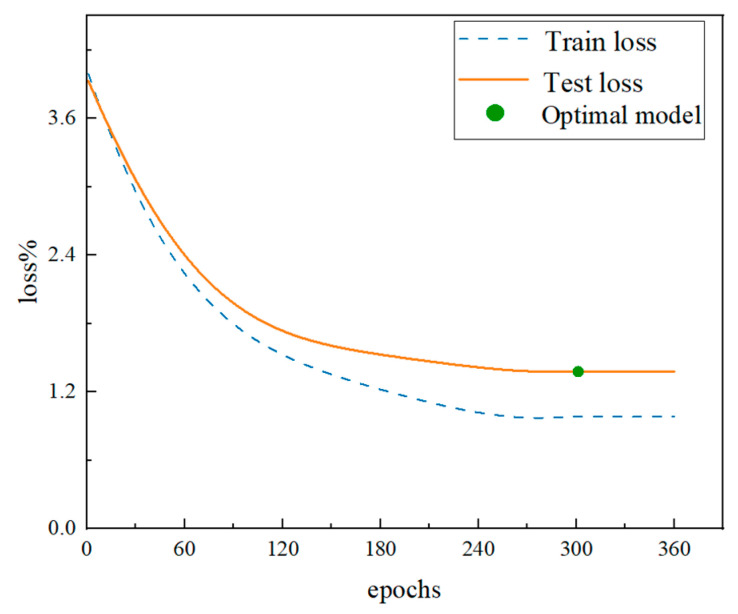
The loss curve of the CNN-TL model.

**Figure 9 sensors-24-07087-f009:**
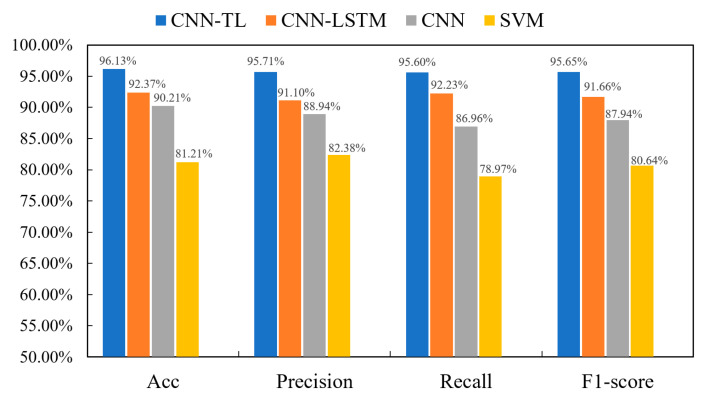
Comparison of the performance of the four classification models.

**Figure 10 sensors-24-07087-f010:**
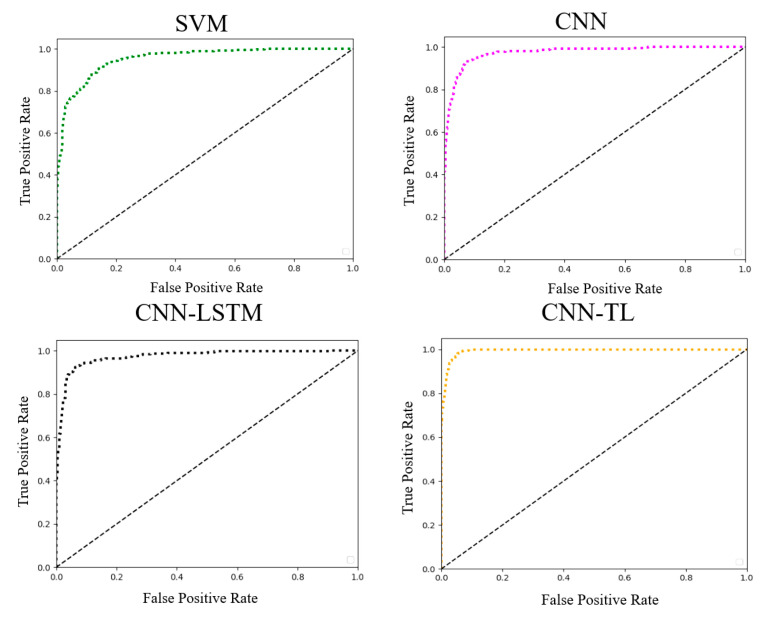
ROC curves for four classification models.

**Figure 11 sensors-24-07087-f011:**
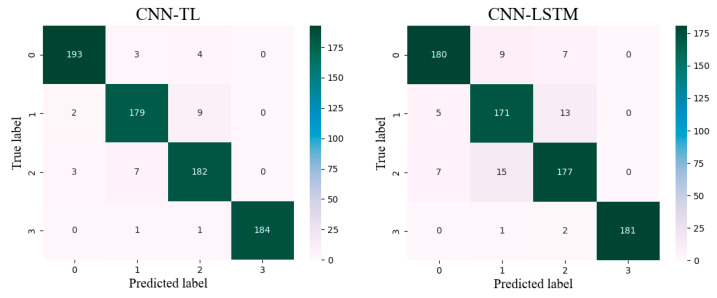
Confusion matrices for four models.

**Table 1 sensors-24-07087-t001:** Subjects’ basic information.

People	Age	Height/(cm)	Weight/(kg)
20	22~28	160~189	60~85

**Table 2 sensors-24-07087-t002:** Muscle, electrode placement, recording channels.

Muscle Name	Patch Position	Acquisition Channel
Rectus femoris	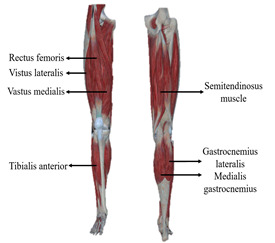	EMG-ch1
Vastus lateralis		EMG-ch2
Vastus medialis		EMG-ch3
Semitendinosus muscle		EMG-ch4
Tibialis anterior		EMG-ch5
Gastrocnemius lateralis		EMG-ch6
Medial gastrocnemius		EMG-ch7

**Table 3 sensors-24-07087-t003:** Feature calculation formula.

	Feature Value Name	Formula of Calculation
Time-Domain Features	MAV	$T_{MAV}=\frac{1}{N}\sum_{i=1}^{N}\left|x_{i}\right|$
	RMS	$T_{RMS}=\sqrt{\frac{1}{N}\sum_{i=1}^{N}x_{i}^{2}}$
Frequency- Domain Features	MPF	$F_{MPF}=\int_{0}^{+\infty }fP(f)df/\int_{0}^{+\infty }P(f)df$
	MF	$\int_{0}^{F_{MF}}P\left(f\right)df=\frac{1}{2}\int_{0}^{+\infty }P(f)df$

**Table 4 sensors-24-07087-t004:** 1DCNN model parameter settings.

Network Layer	Kernel Size	Kernels Number	Output
Convolutional layer 1	3 × 1	64	64 × 1020
BN1+ReLU	64	-	64 × 1020
Max-Pool 1	2 × 2	64	64 × 510
Convolutional layer 2	3 × 1	32	32 × 508
BN2+ReLU	32	-	32 × 508
Max-Pool 2	2 × 2	32	32 × 254
Convolutional layer 3	3 × 1	10	10 × 252
BN3+ReLU	10	-	10 × 252
Max-Pool 3	2 × 2	10	10 × 126
Fully connected layer 1	1260	1	1260 × 1
Fully connected layer 2	600	1	600 × 1
Fully connected layer 2	100	1	100 × 1

**Table 5 sensors-24-07087-t005:** LSTM model parameter settings.

**Hyperparameters**	**Value**
loss function	Cross Entropy
optimizer	Adadelta
layers	3
LSTM unit	100
batch size	40
learning rate	0.001

**Table 6 sensors-24-07087-t006:** Comparison of model results (%).

Model	Acc	Pre	Rec	F1
SVM	81.21	82.38	78.97	80.64
CNN	90.21	88.94	86.96	87.94
LSTM	84.03	86.64	82.37	84.45
CNN-LSTM	92.37	91.10	92.23	91.66
CNN-Transformer	92.06	91.28	91.14	91.26
CNN-TL	96.13	95.71	95.60	95.65

## Data Availability

The original contributions presented in this study are included in the article. Further inquiries can be directed to the corresponding author.

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
