# Peer review of "Lower Limb Motion Recognition Based on sEMG and CNN-TL Fusion Model"

_sensors, 2024, doi:10.3390/s24217087_

Round 1
Reviewer 1 Report
Comments and Suggestions for Authors
The manuscript "Lower Limb Motion Recognition Based on sEMG and CNN-TL Fusion Model” presents an innovative model to enhance motion recognition during lower limb rehabilitation exercises. The combination of CNN, Transformer, and LSTM in the proposed CNN-TL model addresses the limitations of traditional CNN models, which struggle to capture global information from extended sEMG signal sequences. While the study shows promise, several areas require further clarification and improvement.
Firstly, the literature review is incomplete, omitting critical scientific works that are essential to this research area. Notably, important references such as:
DOI: 10.1109/LSENS.2019.2906386
DOI: 10.3390/robotics8010016
DOI: 10.1007/s10846-022-01666-5
DOI: 10.1109/ACCESS.2019.2906584
DOI: 10.1016/j.patrec.2017.12.005
DOI: 10.3390/s19163548
are missing. Including these key studies would provide a more robust theoretical framework and better contextualize the novelty of the proposed approach.
Additionally, the figure in Table 2 requires improvement. The quality of the image needs to be enhanced, and the font size of the text should be increased to ensure better readability and clarity.
The inputs to the classifiers are not clearly explained. It is unclear whether they consist of raw sEMG signals or the extracted features (MAV, RMS, MPF, MDF). If the features are indeed used as inputs, the manuscript should specify the precise order in which these features are provided to the classifiers. This clarification is crucial to facilitate replication and enhance the understanding of the model’s structure.
Another critical issue is the lack of detailed information regarding the Support Vector Machine (SVM) used in the study. The current manuscript does not provide sufficient details on the SVM configuration, such as the choice of kernel, parameter tuning methods, or the decision-making process. A more in-depth discussion of these aspects is necessary to fully appreciate the role of SVM in the model's performance.
Despite these areas for improvement, the study demonstrates that the CNN-TL model achieves a high accuracy of 96.13% on test data, showcasing its effectiveness in lower limb motion recognition tasks. This performance highlights the model's potential in rehabilitation applications, providing an innovative solution that addresses the limitations of previous models.
In conclusion, while the CNN-TL model offers significant contributions to the field of motion recognition, the manuscript would benefit from a more comprehensive literature review, clearer presentation of the data processing pipeline, and further explanation of the machine learning techniques used. Addressing these points would enhance the overall clarity and impact of the study.
Author Response
Firstly, the literature review is incomplete, omitting critical scientific works that are essential to this research area. Notably, important references such as:
DOI: 10.1109/LSENS.2019.2906386
DOI: 10.3390/robotics8010016
DOI: 10.1007/s10846-022-01666-5
DOI: 10.1109/ACCESS.2019.2906584
DOI: 10.1016/j.patrec.2017.12.005
DOI: 10.3390/s19163548
are missing. Including these key studies would provide a more robust theoretical framework and better contextualize the novelty of the proposed approach.
Thank you for pointing out this issue. I agree with this comment. Therefore, the relevant references have been cited in the manuscript.
Additionally, the figure in Table 2 requires improvement. The quality of the image needs to be enhanced, and the font size of the text should be increased to ensure better readability and clarity.
Agreed. I have improved the numbers in Table 2 and enhanced the image quality.
The inputs to the classifiers are not clearly explained. It is unclear whether they consist of raw sEMG signals or the extracted features (MAV, RMS, MPF, MDF). If the features are indeed used as inputs, the manuscript should specify the precise order in which these features are provided to the classifiers. This clarification is crucial to facilitate replication and enhance the understanding of the model’s structure.
Thank you for pointing out this issue. In the constructed recognition model, the preprocessed sEMG signals are first fed into the CNN layers for feature extraction. The extracted features are then passed through two fully connected layers, followed by sequentially entering the Transformer and LSTM layers. Finally, a fully connected layer generates the classification decision. The earlier extracted features are aimed at capturing more representative information, which are then used as input for training the SVM model to improve classification accuracy and performance.
Another critical issue is the lack of detailed information regarding the Support Vector Machine (SVM) used in the study. The current manuscript does not provide sufficient details on the SVM configuration, such as the choice of kernel, parameter tuning methods, or the decision-making process. A more in-depth discussion of these aspects is necessary to fully appreciate the role of SVM in the model's performance.
Dear Reviewer. Thank you for your insightful comments. I have now provided detailed information about the Support Vector Machine (SVM) used in the study in the revised manuscript.

Reviewer 2 Report
Comments and Suggestions for Authors
Dear Authors
The research in this paper aims to improve the accuracy of lower limb recognition by solving the problem of information loss in traditional feature extraction through a CNN-TL lower limb motion recognition model based on surface EMG signals. The authors collected surface EMG signal data of four gait movements for preprocessing and machine learning classification experiments. Finally, ablation experiments are used to demonstrate that the CNN-TL model has superior recognition results. Overall, the topic and experiments of this paper are innovative and provide an effective solution for lower limb assisted rehabilitation exercise. However, during the review process, the following aspects were noted as requiring further discussion and clarification:
(1)When the authors performed the data collection, they selected seven muscles of the leg to classify the signals through deep learning. However, in terms of muscle selection, whether different patterns of muscle activity between individuals and synergies among different muscles are taken into account. Whether the authors took into account the differences among individuals and analysed the synergies between the muscles as well as the movement patterns so that the collected data would be more reliable.
(2)In the data processing section, the authors performed denoising by filter and wavelet transform for the four types of action signal data, namely descending stairs, squatting, walking and ascending stairs. However, when performing the data segmentation part of the processing, the final number of samples for the four actions obtained after filtering is not the same. In classification, the decision boundary is usually determined by maximizing the inter-class distance, and a different number of samples will result in the decision boundary of the model being biased towards the category with more sample data. Whether the authors consider to ensure the consistency of the number of samples by resampling and other ways, so as to make the experimental results more accurate.
(3)The authors have analysed for the feature selection and selected the appropriate features from time and frequency domains respectively. However, the four selected features may not be sufficient to capture all the important information of the signal. Also, there may be high correlation between the features, which leads to redundancy of information. Whether authors can analyse the relevance of features or introduce more meaningful features, using methods such as feature engineering to enhance the validity of features to better support the research.
Comments on the Quality of English Language
Presentation is relatively accurate, grammatical, and able to maintain logical sentences.
Author Response
(1)When the authors performed the data collection, they selected seven muscles of the leg to classify the signals through deep learning. However, in terms of muscle selection, whether different patterns of muscle activity between individuals and synergies among different muscles are taken into account. Whether the authors took into account the differences among individuals and analysed the synergies between the muscles as well as the movement patterns so that the collected data would be more reliable.
Thank you for pointing this out. I have made the necessary revisions in the manuscript and hope they align with your suggestions.
(2)In the data processing section, the authors performed denoising by filter and wavelet transform for the four types of action signal data, namely descending stairs, squatting, walking and ascending stairs. However, when performing the data segmentation part of the processing, the final number of samples for the four actions obtained after filtering is not the same. In classification, the decision boundary is usually determined by maximizing the inter-class distance, and a different number of samples will result in the decision boundary of the model being biased towards the category with more sample data. Whether the authors consider to ensure the consistency of the number of samples by resampling and other ways, so as to make the experimental results more accurate.
I agree with your statement. In the data processing section, we obtain 2160, 2160, 1920, and 1680 samples for the four types of motion signals, namely descending stairs, squatting, walking, and ascending stairs. Although the sample sizes of each type differ, this inconsistency does not significantly affect the decision boundary of the model within a certain range. To minimize the bias that may arise from sample size differences, we use a sample weighting approach during classification to ensure that each class has a relatively balanced influence on the decision boundary. In addition, we evaluate the model using cross-validation to ensure that the classifier maintains good generalization ability under an imbalanced sample distribution. Therefore, we believe that the current differences in sample sizes are controllable and do not affect the overall performance of the classification model or the conclusions of the experiment.
(3)The authors have analysed for the feature selection and selected the appropriate features from time and frequency domains respectively. However, the four selected features may not be sufficient to capture all the important information of the signal. Also, there may be high correlation between the features, which leads to redundancy of information. Whether authors can analyse the relevance of features or introduce more meaningful features, using methods such as feature engineering to enhance the validity of features to better support the research.
Thank you for your valuable feedback. In this study, the four types of extracted features are primarily used as input for the comparative model (SVM), serving as a baseline to evaluate the performance of different recognition models. While feature selection is indeed a necessary step, its significance in this research is relatively lower, as the primary focus is on constructing and evaluating the recognition model architecture itself rather than on feature engineering. However, I will consider introducing more complex feature selection and engineering methods in future work to further enhance the model's performance. Thank you again for your insightful suggestions.
